# A Novel Approach for Efficient Solar Panel Fault Classification Using Coupled UDenseNet

**DOI:** 10.3390/s23104918

**Published:** 2023-05-19

**Authors:** Radityo Fajar Pamungkas, Ida Bagus Krishna Yoga Utama, Yeong Min Jang

**Affiliations:** Department of Electronics Engineering, Kookmin University, Seoul 02707, Republic of Korea; radityofajar@gmail.com (R.F.P.); idabaguskrishnayogautama@gmail.com (I.B.K.Y.U.)

**Keywords:** coupled UDenseNet, aerial thermography, fault classification, GAN

## Abstract

Photovoltaic (PV) systems have immense potential to generate clean energy, and their adoption has grown significantly in recent years. A PV fault is a condition of a PV module that is unable to produce optimal power due to environmental factors, such as shading, hot spots, cracks, and other defects. The occurrence of faults in PV systems can present safety risks, shorten system lifespans, and result in waste. Therefore, this paper discusses the importance of accurately classifying faults in PV systems to maintain optimal operating efficiency, thereby increasing the financial return. Previous studies in this area have largely relied on deep learning models, such as transfer learning, with high computational requirements, which are limited by their inability to handle complex image features and unbalanced datasets. The proposed lightweight coupled UdenseNet model shows significant improvements for PV fault classification compared to previous studies, achieving an accuracy of 99.39%, 96.65%, and 95.72% for 2-class, 11-class, and 12-class output, respectively, while also demonstrating greater efficiency in terms of parameter counts, which is particularly important for real-time analysis of large-scale solar farms. Furthermore, geometric transformation and generative adversarial networks (GAN) image augmentation techniques improved the model’s performance on unbalanced datasets.

## 1. Introduction

Most countries and industries have recently begun to evaluate their energy policies to assist sustainable development by aiming for a net-zero or carbon-neutral future [1,2]. To address the issue of the depletion of fossil fuels and climate change, renewable energy sources (RESs) have begun to garner significant attention across the globe in order to reduce CO${}_{2}$ gas emissions and address the climate change issue by increasing the share of renewables in the energy mix and total final energy consumption (TFEC). Transitioning to low carbon and renewable energy sources is critical for meeting electrical power demands for ecologically friendly and sustainable energy production. Due to the advancement of nanomaterials technology, PV systems are one of the most promising and clean RES types. PV systems have advantages such as being sustainable, having zero noise operation, and having minimal installation fees, making them suitable for a large or small-scale distributed generation (DG) [3].

Photovoltaic (PV) technology converts the sun’s irradiance into electrical energy. It makes use of components such as silicon, which allows for the generation of an electrical current by absorbing photons from sunlight and releasing electrons. PV modules are the primary power-producing component of a PV system, and the efficacy and dependability of PV modules continue to be potential issues due to failures and degradation in the field. In PV facilities, different anomalies affecting operation systems typically cause the source of energy output losses. These defects decrease efficiency and create potential electrical hazards for PV system operators. PV modules are subjected to various environmental stresses and extreme circumstances that risk their dependability and durability throughout their lives. This damage could lead to abnormal operation, safety issues, and fire hazards, reducing PV modules’ lifespan [4,5].

The volume of PV module waste is increasing rapidly as the use of PV systems increases. Furthermore, PV module waste, mostly made of crystalline silicon (c-Si) material, could pollute the environment with heavy metals that are difficult to extract during recycling [6]. Furthermore, it is acknowledged that thorough inspection and maintenance of PV systems are required to maintain optimal performance. Identifying the economically optimal interval for inspection and maintenance interventions is critical. In order to find and fix module failures in time to extend their lifetime and maintain the system’s optimal operating efficiency, which will lead to higher financial return, it is crucial to develop methods for accurately detecting and classifying defects in PV systems in a rapid, accurate, and efficient manner.

For fault detection in PV modules and cells, electroluminescence (EL) imaging, infrared (IR) imaging, and electrical measurement and characterization are extensively utilized approaches. Electrical assessment involves assessing the output performance of PV panels by measuring the current–voltage (I-V) curve. On a synthetic dataset, the combination of electrical assessment and artificial intelligence can achieve a classification error of 2.7% [7]. However, it cannot pinpoint the location of flaws and requires exact illumination and temperature [8]. On the other hand, visual and thermal methods such as IR and EL can pinpoint the exact location of defects. However, EL imaging that requires a special environment is impractical for large-scale outdoor applications [9], and conventional IR imaging is time-consuming for large-scale solar farms. Furthermore, these methods necessitate the assistance of a thermography inspection expert to assess and verify the issue [9,10].

Aerial thermography, or unmanned aerial vehicle inspection, has developed as a more efficient, dependable, and cost-effective alternative to conventional visual monitoring for detecting faults and failures in photovoltaic (PV) modules, particularly in high-risk human field zones [11]. However, correctly diagnosing defects in aerial thermography images is difficult. Deep learning models, such as CNNs, are one challenge that necessitates enormous volumes of annotated data and is both time-consuming and costly to implement. Another challenge is establishing high accuracy in fault classification, which can be affected by environmental factors and PV module variability. Lightweight models are also essential for real-time analysis of aerial thermography images and for reducing processing expenses. As a result, overcoming these obstacles is critical for successfully adopting aerial thermography in PV fault detection and classification.

We propose a novel hybrid approach combining the coupled U-Net architecture and the dense block from DenseNet to achieve accurate and significantly lightweight PV fault classification performance. Our novelty and main contributions to this paper are as follows:The proposed hybrid approach combining the coupled U-Net architecture and DenseNet dense blocks enables uninterrupted gradient flow, feature reusability, and training stabilization, resulting in accurate PV fault classification performance that exceeds similar studies on the same dataset.The proposed coupled UDenseNet model performs thorough classification of 2-class (Fault/No-fault), 11 types of faults, and 12 types of PV conditions, which have been validated across 826 real-world solar PV installations across six continents, significantly boosting the model’s generalization capability, and these 11 types of PV faults are introduced in Table 1.To improve accuracy on imbalanced data, the presented model is trained using geometric transformation and GAN image augmentation techniques in conjunction with oversampling methods, further improving fault classification accuracy in aerial thermography images and allowing for more effective implementation in PV fault detection and classification.

The rest of this paper is structured as follows: Section 2 provides an overview of the related works for PV fault classification using a publicly available dataset [10]. Section 3 provides a complete explanatory analysis of the dataset and data preprocessing. Section 4 provides an outline of our proposed strategy as well as how we run the experiments. The findings of our trials and comparisons with other methodologies are presented in Section 5. Finally, Section 6 summarizes our findings and discusses future work.

## 2. Literature Review

Classification and evaluation of observed defects in solar panels necessitate an in-depth understanding of solar technology as well as knowledge of the inspected system. Various advanced fault detection and diagnostic (FDD) approaches for classifying PV panel problems have been presented in recent years. Deep learning-based approaches for detecting and classifying anomalies in thermographic PV images have become more popular as machine learning has advanced. Deep learning algorithms extract and learn features more effectively, resulting in more accurate and robust classification performance.

On the other hand, deep learning algorithms often demand a massive quantity of data, and examining thermal images of solar modules requires the expertise of an expert to spot anomalies and label the data. As a result, data availability remains a challenge for machine learning researchers. Millendorf et al. [10] provide a publicly available dataset that includes real images of 11 class anomalies. This dataset has been used in several studies on PV fault classification.

For example, Le et al. [12] offered an ensemble of different ResNet-based structure models with varied sets of data augmentation and minority class increment to obtain an average accuracy of 94% in binary classification and 85.9% in multi-class classification of 12 fault types. The study also looked at the effects of data augmentation, oversampling, SMOTE, and focal loss on the unbalanced dataset, which led to 2.9% and 7.4% improvements for the 2-class and 12-class outputs, respectively. Another work, Fonseca Alves et al. [13], classified 11 different anomaly classes using a CNN-based model paired with undersampling and oversampling approaches on an unbalanced dataset. Through cross-validation, this technique attained an accuracy of 92.5% in binary classification, 66.43% in classifying 12 fault types, and 78.85% in classifying faults for eight selected classes.

Similarly, Korkmaz et al. [14] suggested a multi-scale CNN with three branches based on pre-trained AlexNet architecture and an offline augmentation approach for classifying 11 different anomaly categories. The authors increased the input image size to 227 × 227 pixels and the total model parameter to around 42M. This approach had an average accuracy of 97.32% for two-class outputs and 93.51% for 11 anomalous class outputs. These studies show that various approaches, such as data augmentation, oversampling, and pre-trained models, can significantly improve the performance of image classification algorithms on imbalanced datasets.

## 3. Dataset Preparation

Long-term operation of solar PV panels can expose them to a wide variety of potential faults. There is a lack of publicly available datasets that provide aerial thermographic images of various anomalies in PV systems as a result of the fact that anomalies are rare, and this method needs to be analyzed and labeled by an expert. This is because anomalies require an expert to perform classification tasks. Therefore, an original and widely accessible dataset called the Infrared Solar Modules dataset, which is licenced under an MIT copyright and contains aerial thermographic images of numerous PV anomalies identified in practical solar power plants, was chosen in order to provide an accurate classifier architecture.

The dataset’s collecting, processing, labelling, and categorization were all handled by the Raptor Maps team. They completed this work using standardised inspection methods, including solar panel infrared imaging [15]. All of the images were captured using visible spectrum cameras integrated into an unmanned aerial vehicle (UAV) system or piloted aircraft, along with midwave or longwave infrared (3–13.5 µm) cameras in a grayscale colorspace. The discovered abnormalities were divided into groups based on the structure of the classes and cropped to each specific module. The data were collected in 2019 from 25 countries and a total of 826 solar PV systems across 6 continents.

The dataset contains 20,000 images, with temperature values represented by 24 × 40 × 1 pixels per image. Due to the varying distance between the UAV and the PV modules, the spatial resolution of the images ranges from 3.0 to 15.0 cm/pixel. There are 12 different classes, consisting of 1 normal module class and 11 anomaly classes (cell, hotspot, cracking, diode, shadowing, etc.). Table 1 displays the comprehensive description and random samples for each class. Figure 1 provides a graphical representation of data distribution.

Based on the total number of existing global findings, the proportion of classes in the dataset is unbalanced (e.g., the fault class cell has 1877 images and diode multi class only has 175 images). The unbalance of classes within the dataset poses a significant issue for the machine learning-based classification method. In addition, although some anomalies were easy to recognize and classify, others were considerably more complicated to differentiate. Consequently, it is essential to develop a deep-learning model that can automatically detect and classify panel anomalies without the assistance of an expert.

Unbalanced class distributions in datasets can substantially affect the performance of image classification models in deep learning because unbalanced class proportions can impact the neural network training and its ability to generalize to unknown examples. A common solution to this problem is oversampling or undersampling the original images in the dataset [16].

Undersampling means decreasing the amount of majority-class data to match the amount of minority-class data. This approach can be accomplished by arbitrarily removing samples from the majority class or by employing algorithms designed to select a subset of the majority class that most accurately represents it. However, this method can result in the loss of information, making it more difficult for the model to discover the underlying data pattern.

In contrast, oversampling entails adding additional samples to the minority group. Using techniques such as synthetic minority over-sampling technique (SMOTE) [17], adaptive synthetic sampling (ADASYN) [18], GAN [19], and variable autoencoders (VAE) [20], it is possible to generate synthetic samples for this method. This method has the benefit of increasing the number of samples for the minority class, making it easier for the model to discover the underlying data pattern. However, it is not guaranteed that synthetic data will have the same characteristics as the original data, particularly in complex and high-dimensional datasets [21].

In this study, we propose an oversampling method based on image augmentation through geometric transformations to increase the total amount of images of the minority classes to match the majority classes while preserving the pattern of the fault. The transformations used include horizontal and vertical flipping, shifting, and adjusting brightness. Furthermore, we employed and analyzed GAN image augmentation techniques for the oversampling [22].

Structural similarity index (SSIM) was used as metric to quantify the similarity between real images and GAN-augmented images [23]. Based on Table 2, it can be observed that for most anomaly classes, the maximum SSIM score of GAN-augmented images is lower than that of raw data. This suggests that the GAN augmentation technique may introduce some level of distortion or dissimilarity in the images, particularly for subtle anomalies such as cracking and vegetation. However, mean SSIM scores for most anomaly classes are higher for GAN-augmented images than for raw data, indicating that the overall image quality is improved by the GAN technique. This improvement in the mean SSIM score can be attributed to the fact that GANs generate new images that are visually similar to the original ones but have some level of variation, which can enhance the diversity and richness of the dataset. However, it is important to note that the improvement in the mean SSIM score is not uniform across all anomaly classes and that the GAN technique may not be suitable for all types of image data.

Following the data augmentation and oversampling processes, the dataset consisted of 88,000 images for 2-class experiments, 110,000 images for 11-class experiments, and 120,000 images for 12-class experiments. The images were balanced across the different fault classes and had a resolution of 24 × 40 pixels. We split the data into training, validation, and testing sets with distributions as shown in Table 3. The 2-class experiments or binary classification (error or no error) are beneficial when the main goal is to ascertain whether or not the PV module is defective. When creating, setting up, or managing PV systems, this classification is frequently used to ensure quality. The 11-class experiment is useful when the main goal is to identify a specific type of fault that is present in the PV module. This classification can assist the maintenance technicians in identifying and fixing issues that might affect system performance. Furthermore, the 12-class experiment is useful when the purpose is to categorize a variety of PV module situations, including normal and all defective conditions.

## 4. Proposed Method

In this study, we propose a novel architecture, Coupled UDenseNet, that combines the advantages of the UNet and DenseNet architectures to improve image classification performance for the anomaly in PV panels. The coupled UDenseNet model is composed of two UDenseNet that are positioned in a series and connected with a coupled connection to make information flow more efficient across UNet. The UNet architecture is chosen for its effectiveness in feature extraction [24], whereas DenseNet is known to promote feature reuse and better gradient flow during training [25]. By combining these two architectures, the authors aim to leverage the advantages of both to improve performance for this specific task. In the original UNet, each multi-channel feature map applied two 3 × 3 convolutions, followed by a ReLU and a 2 × 2 max pooling operation.

To improve the information flow in the UNet, we propose to utilize DenseNet in the multi-channel feature map of the UNet. Thus, in this work, we called the multi-channel feature map as a dense block. The dense block is an important component of the coupled UDenseNet architecture. It connects each layer in a feedforward fashion, resulting in enhanced feature propagation and reuse. In particular, the output of each layer is concatenated with the input of the following layer, allowing the network to learn more complex features [25]. Mathematically, the dense block can be expressed as:(1)$$x_{l}=H_{l}\left(\left[x_{0},x_{1},\cdots ,x_{l}\right]\right)$$
where *l* represents the *l*th layer on the dense block, $\left[x_{0},x_{1},\cdots ,x_{l}\right]$ refers to the concatenation in the *l*th layer, and $H_{l}$ is the composite function of convolution, LeakyReLU, and BatchNorm.

Figure 2 depicts the coupled UDenseNet architecture. Skip connections are used to connect components of each UDenseNet in order to create a coupled connection between two UDenseNets [26,27]. Both UDenseNets have the same architecture, which performs downsamples and upsamples three times. Each downsample’s image size is reduced by half by performing a dense block and pooling layer. Following the downsampling, the image is upsampled to its original size.

The coupled connection enables a block to receive features from the same block in the preceding UNet. Hence, each block in the coupled UDenseNet receives two features from the previous block of the same UNet and the same block in the preceding UNet. The purpose of coupling connections between two UNets is to improve the gradient flow to the later layers. Therefore, the learning performance can be improved, and it is possible to achieve higher accuracy in classification performance. The mathematical model of the coupled UDenseNet can be represented as:(2)$$q_{a}^{b}={x_{l}}^{\prime }\left(\left[q_{a-1}^{b},Q_{a}^{b-1}\right]\right)$$
(3)$$w_{a-1}^{b}={x_{l}}^{\prime \prime }\left(\left[w_{a}^{b},q_{a}^{b},W_{a}^{b-1}\right]\right)$$
(4)$$Q_{a}^{b-1}=q_{a}^{b-k},\cdots \to ,q_{a}^{b-1}$$
(5)$$W_{a}^{b-1}=w_{a}^{b-k},\cdots \to ,w_{a}^{b-1}$$
where $Q_{a}^{b-1}=q_{a}^{0},q_{a}^{1},\cdots \to ,q_{a}^{b-1}$ is defined as the outputs of the *a*th or the top-down blocks in UDenseNet. Similarly, $W_{a}^{b-1}=w_{a}^{0},w_{a}^{1},\cdots \to ,w_{a}^{b-1}$ are the outputs of the *b*th or the bottom-up blocks. The ${x_{l}}^{\prime }$ and ${x_{l}}^{\prime \prime }$ in the equation represent operations of the dense block, transpose of convolution, pooling, and spatial dropout. The notation (⋯→) denotes the feature concatenation process, which ensures an uninterrupted flow of information. Additionally, the value of 0≤k represents how many preceding UDenseNet connections are used.

At the end of the coupled UDenseNet, a dense layer is attached, which consists of two neurons for binary classification and 11 or 12 neurons for multi-class classification. The dense layer’s function generates the class result based on the relevant features learned in the coupled UDenseNet. Following the dense layer, an activation function is applied, where the Sigmoid activation function is utilized for binary classification and the Softmax activation function is for multi-class classification.

The detailed components of UDenseNet are shown in Figure 3. The input block is depicted in Figure 3a, which consists of two series of convolution, LeakyReLU, and BatchNorm. The input layer performs initial feature extraction for the input image. Next, the downsampling block is shown in Figure 3b, where there are three downsampling blocks implemented. Each downsampling block performs feature size reduction into half-size by using a series of max pooling and dense blocks.

Next, the downsampling block is followed by three upsampling blocks, shown in Figure 3c. In the upsampling block, a transpose of convolution is utilized to restore the initial feature size. In addition, the output features from the previous block are concatenated with the features from the skip connection to preserve the feature information. To select the useful features and prevent overfitting, spatial dropout is applied in the upsampling layer.

Subsequently, the features are fed to the second downsampling block of the coupled UDenseNet, which is shown in Figure 3d. The principle in the second downsampling block is similar to that in the previous downsampling block. The difference is only in the use of spatial dropout on the end of the block to prevent overfitting. The second upsampling block is applied after the second downsampling block and presented in Figure 3e. Again, the second upsampling block is similar to the previous upsampling layer, but in this block, spatial dropout is not utilized. All downsampling and upsampling blocks in the second UDenseNet receive additional features from the preceding UDenseNet.

Finally, after the second upsampling block, the classification block is applied to generate the class prediction. The architecture of the classification block is depicted in Figure 3f, where it consists of additional convolution and BatchNorm. A flatten layer is utilized to map the 2-dimensional features from the preceding blocks into 1-dimensional features. The flatten layer is important because we use the dense layer to generate the class prediction, and the dense layer cannot use 2-dimensional features as input. After flattening the features, they are fed to the dense layer, which includes a Sigmoid activation function or a Softmax activation function, depending on the classification purpose.

## 5. Experiments

Experimental investigations are conducted in this study to evaluate the proposed method. The experiments were performed in Python 3.10.6 with an AMD Ryzen 5 3400G CPU, an NVIDIA RTX 3080Ti GPU, and 24 GB of RAM. The Keras 2.10.0 library, which runs on the TensorFlow 2.10.0 framework, was employed for training and simulation. Three sets of tests were carried out: a 2-class classification of anomaly and no-anomaly, 11 different types of PV faults, and a 12-types of PV conditions, encompassing no-anomaly and 11 different anomaly classes in total.

The deep learning model was trained with the AdamW [28] optimizer, 32 batch size, and a learning rate of 0.001, with a learning rate decay introduced if the validation loss did not decrease within ten epochs. The maximum number of epochs was set at 200, and the early stopping strategy was employed to avoid overfitting. Table 4 describes a complete hyperparameter tuning used to train the coupled UDenseNet model.

### 5.1. Evaluation Metrics

The performance of the proposed coupled UDenseNet model is evaluated using a variety of evaluation indicators. Accuracy is a popular image classification evaluation metric that evaluates the total correctness of the model’s predictions. It is measured as the proportion of correct estimations to total predictions. In image classification, accuracy is defined as the proportion of correctly identified images. It is important to note, however, that accuracy alone may not provide a complete representation of the model’s performance, especially when the dataset is unbalanced.

To address this issue, precision and recall are frequently employed in concert with accuracy. Precision in image classification is a proportion of true classified positive images among all positive images, whereas recall is the proportion of true classified positive images among all actual positive images in the dataset. The F1 score, which is the harmonic mean of precision and recall, is another extensively used image categorization evaluation metric. It provides a balanced measure of precision and recall, which is especially beneficial when the dataset is unbalanced.

To construct these evaluation measures, a confusion matrix is often employed, which indicates the number of correct and incorrect predictions made by the model for each class in the dataset. The following formulas can be used to determine the accuracy, precision, recall, and F1 score from the confusion matrix:(6)$$\mathrm{Accuracy}=\frac{\mathrm{TP}+\mathrm{TN}}{\mathrm{TP}+\mathrm{FP}+\mathrm{TN}+\mathrm{FN}}$$
(7)$$\mathrm{Precision}=\frac{\mathrm{TP}}{\mathrm{TP}+\mathrm{FP}}$$
(8)$$\mathrm{Recall}=\frac{\mathrm{TP}}{\mathrm{TP}+\mathrm{FN}}$$
(9)$$F1\;\mathrm{score}=\frac{2\mathrm{TP}}{\left(2\mathrm{TP}+\mathrm{FP}+\mathrm{FN}\right)}$$
where TP, FP, TN, and FN are the number of true positives, false positives, true negatives, and false negatives, respectively.

To analyze the model efficiency, we use model parameters as the evaluation metric. The total parameter refers to the number of adjustable parameters in a model that can be learned from data during training. These parameters include the weights and biases of the model’s layers, which are adjusted to its performance. Additionally, the parameter count can also impact the model’s computational efficiency, particularly for large-scale applications, making it an important consideration for practical implementation.

### 5.2. Result and Discussion

The first simulation will be conducted to ascertain the presence of any defect in a specific solar panel. In the second and third simulation scenarios, the images labeled as anomalies are segregated based on their respective anomaly categories.

#### 5.2.1. The First Case: 2-Class Output

The evaluation of the proposed model’s performance was conducted utilizing diverse data augmentation methodologies and distributions. Figure 4 displays the validation accuracy and loss of the model during training. The results indicate that the validation loss and accuracy stabilized after approximately 60 epochs. The study’s findings revealed that the utilization of geometric transformation and GAN augmentation methodologies had a remarkably positive impact on the model’s performance. Although the impact of the data distribution was minimal, it was noted that the proposed model demonstrated greater efficacy in capturing the fundamental patterns inherent in the data.

The findings of the 2-class classification experiment are presented in Table 5. The results indicate that the raw data, without augmentation, attained an accuracy of 92.22% on the test dataset, using a data split of 70% for training, 20% for validation, and 10% for testing. Using geometric transformation alone or combined with GAN enhanced accuracy rates to 99.17% and 97.36% on the test dataset, respectively, for the identical data split. The observed enhancement shows that the proposed approach can extract more useful information from the augmented data than the raw data.

The results indicate a positive correlation between the proportion of data utilized for training and the overall enhancement of the model’s performance. Notably, the maximum training data did not necessarily yield the highest values for accuracy, precision, recall, and F1 score. The study found that the most favorable equilibrium between the quantity of training data and the model’s capacity to generalize was attained by utilizing a data split of 75% for training, 20% for validation, and 5% for testing across all augmentation methodologies. Furthermore, it was observed that the geometric transformation technique’s sole employment yielded superior results compared to the utilization of the geometric transformation with the GAN technique. The augmented complexity introduced by the GAN technique may not be necessary for this specific scenario. Moreover, the confusion matrix for 12-class output is presented in Figure 5.

#### 5.2.2. The Second Case: 11-Class Output

Figure 6 depicts the validation accuracy and loss trends for the proposed technique during training for the 11-class output case. The validation loss stabilized after approximately 70 epochs. The data split of 80% train, 10% validation, and 10% test exhibited the most fluctuation of all data augmentations, showing that this distribution influenced the model’s performance. The results of the picture classification task with 11 output classes utilizing various data augmentation strategies and distributions are presented in Table 6. Raw data had relatively poor accuracy compared to other augmentation strategies, ranging from 64% to 67.9% for all data split distribution on the test dataset. On the other hand, the geometric transformation produced significantly higher accuracy, ranging from 94.43% to 96.65% for all data distribution on the test dataset, proving its ability to improve the model’s performance.

On the other hand, the efficiency of integrating geometric transformation with GAN varies depending on the data distribution. For example, while utilizing a data split of 75% train, 20% validation, and 5% test, the accuracy of geometric transformation combined with GAN was 94.22% on the test dataset, outperforming raw data but falling short of geometric transformation in identical data split distribution. Although GANs can produce realistic images, they can also produce false positives or negatives, resulting in incorrect classifications. Furthermore, the number of output classes in this task is significantly higher than in the previous task, which had only two output classes, making it more difficult as well as explaining the raw data’s lower accuracy. Moreover, the confusion matrix for 11-class output is presented in Figure 7.

#### 5.2.3. The Third Case: 12-Class Output

Figure 8 illustrates the proposed method’s training performance for the 12-class output case, demonstrating a similar pattern to previous tasks. The validation loss stabilizes at approximately 75 epochs; however, the loss graph fluctuates considerably because of the increased false alarms caused by the no-anomaly class. Table 7 displays the results of the picture classification task with 12 output classes utilizing various data augmentation strategies and data distributions. Consistent with prior examinations, raw data produce low accuracy ranging from 78.6 to 80.4% on the test dataset across all data distributions, highlighting the relevance of data augmentation in enhancing model performance in imbalanced datasets. The geometric transformation alone results in a considerable increase in accuracy ranging from 94.2% to 95.7% on the test dataset, demonstrating its efficiency in boosting model performance for this specific context. The confusion matrix for 12-class output is presented in Figure 9.

Although there may be some numerical variation between these results and previous analyses of image classification tasks with 2 and 11 output classes, the overall trend remains consistent: raw data yield poor performance, the geometric transformation significantly improves accuracy, and the effectiveness of combining geometric transformation with GAN varies depending on the data distribution. This consistency in results emphasizes the need to carefully select data augmentation approaches when training image classification models.

#### 5.2.4. Comparison with the Previous Study

In Table 8, we compare the performance of various models on the same dataset for a 2-class classification task. The CNN model developed by [13] attained 92.5 % accuracy, with precision and recall values of 92.00%. In contrast, the Ensemble model proposed by [12] achieved a higher accuracy of 94.40%, but with no reported precision or recall values. The Transfer Learning and Multiscale CNN model proposed by [14] produced an even higher accuracy of 97.32%, with precision and recall values of 97.63% and 97.00%, respectively.

With an outstanding value of 99.39%, our proposed technique attained the best accuracy of all models. Our model also has a high precision value of 98.79% and an excellent recall value of 100%, for a total F1 score of 99.39%. Our model has 13.9M parameters, many fewer than the 42M parameters employed by the Transfer Learning and Multiscale CNN models. Although it is not the lowest, it implies that it is lightweight and computationally efficient. These findings indicate that our suggested strategy outperforms the other models on the same dataset for this 2-class classification job, suggesting the usefulness of our approach.

Similar to Table 8, Table 9 shows the performance evaluation of different models for the multi-class image classification on the same dataset. The first model, CNN structure by [13], was evaluated on an 11-class output, and its performance is reported with an accuracy of 66.43% for the 11-class output. The second model, Ensemble model by [12], was evaluated on a 12-class output, and its performance is reported with an accuracy of 85.90%. The third model, Transfer Learning and Multiscale CNN by [14], was evaluated on an 11-class output, and its performance is reported with an accuracy of 93.51%.

The proposed method in this study was evaluated on both 11-class and 12-class output and is reported to have an accuracy of 96.65% and 95.72%, respectively. This model was developed using a coupled UDenseNet architecture with 13.9 million parameters. The proposed method outperforms all the other models in accuracy, precision, recall, and F1-score, demonstrating its superiority in multi-class image classification.

## 6. Conclusions

Photovoltaic (PV) systems are eco-friendly, noiseless, and inexpensive to install. Field failures and degradation may decrease PV module reliability and durability. This might produce malfunctions, safety issues, and fire hazards, reducing PV module lifespan. Heavy metal-containing PV module waste is growing. Therefore, PV systems need frequent inspection and maintenance.

The evaluation of the proposed model’s performance was conducted using various data distributions and data augmentation techniques, such as geometric transformation and GAN image augmentation. The results showed that the utilization of geometric transformation and GAN augmentation methodologies had a positive impact on the model’s performance. Furthermore, the effectiveness of combining geometric transformation with GAN varied depending on the data distribution, and the augmented complexity introduced by the GAN technique may not be necessary for this specific scenario.

Our analysis has shown that the accuracy of the proposed model was notably superior compared to previous studies conducted on the identical dataset. The proposed model achieved an accuracy of 99.39%, 96.65%, and 95.72% for 2-class, 11-class, and 12-class output on the test dataset, respectively. Furthermore, the accuracy of our model was 2–3% higher than the best-performing model reported in the literature, which utilized the Transfer Learning and Multiscale CNN approach and possessed more than three times the total parameter counts of our proposed model. These results demonstrate that the proposed approach can significantly improve the accuracy of PV fault detection, which can lead to improved maintenance of PV systems and increased energy efficiency.

In the future, further research can explore the combination of coupled charge device (CCD) cameras and thermal cameras to obtain a more comprehensive diagnosis of PV faults. Additionally, there is a need to investigate the feasibility of implementing the proposed approach in real-world settings. Moreover, there is scope for extending the current model to predict the remaining useful lifetime (RUL) of PV modules to enhance the maintenance schedule of solar PV plants. Finally, there is a need to consider developing a cloud-edge architecture with a user-friendly software tool that can automatically detect and classify faults in real time, thereby minimizing manual inspections and reducing downtime for solar PV plants.

## Figures and Tables

**Figure 1 sensors-23-04918-f001:**
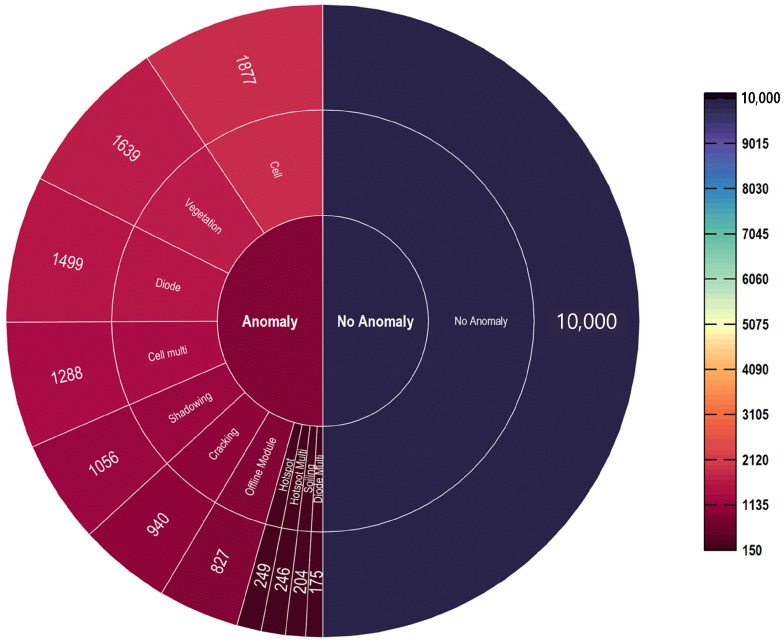
Total images in the dataset for each class.

**Figure 2 sensors-23-04918-f002:**
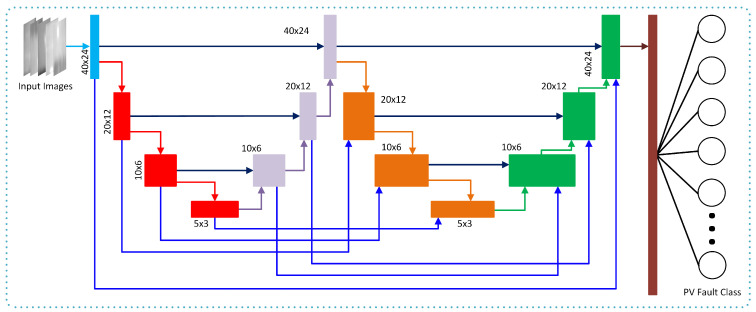
An illustration of the proposed coupled UDenseNet architecture designed for the accurate classification of PV faults. Each color represents specific block and function.

**Figure 3 sensors-23-04918-f003:**
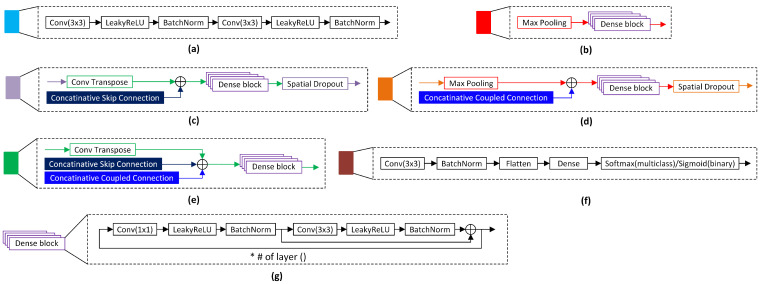
Details on the layer of coupled UDenseNet architecture. (**a**) Input block, (**b**) downsampling block, (**c**) upsampling block, (**d**) downsampling with coupled connection block, (**e**) upsampling with coupled connection block, (**f**) classification block, (**g**) dense block. In a dense block, the asterisk symbol represents the total number of dense blocks used.

**Figure 4 sensors-23-04918-f004:**
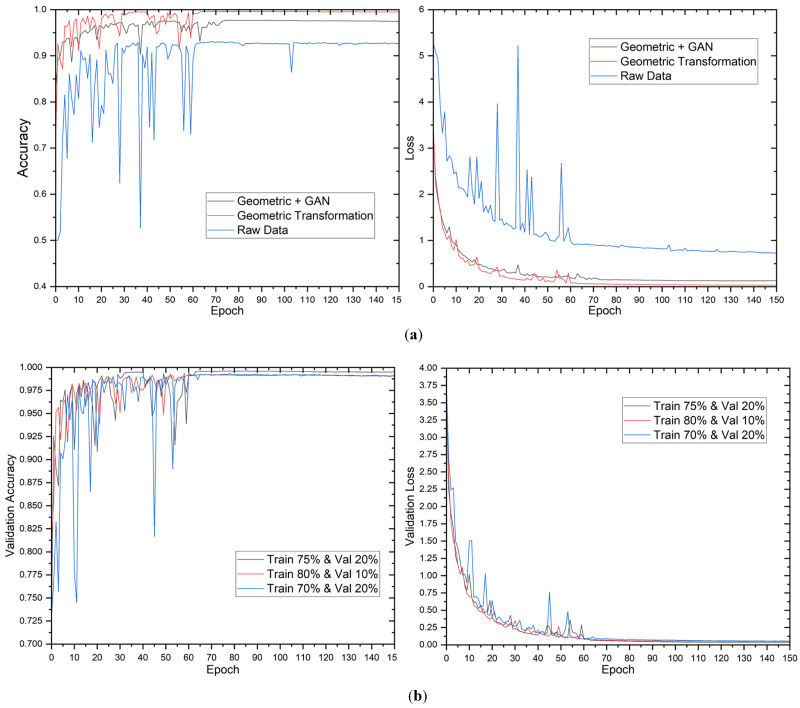
Evolution of validation loss and accuracy for 2-class outputs. Evolution of validation loss and accuracy in different (**a**) data augmentation methods and (**b**) data distributions.

**Figure 5 sensors-23-04918-f005:**
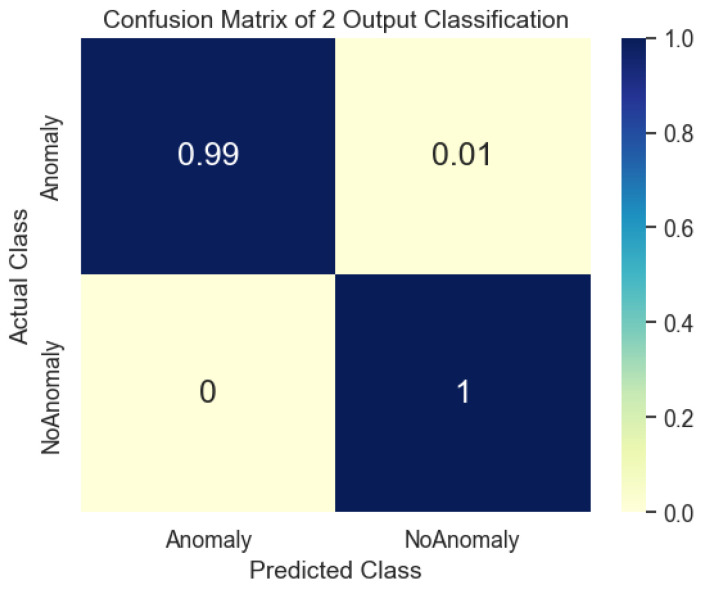
A confusion matrix of the coupled UDenseNet model for 2-class output.

**Figure 6 sensors-23-04918-f006:**
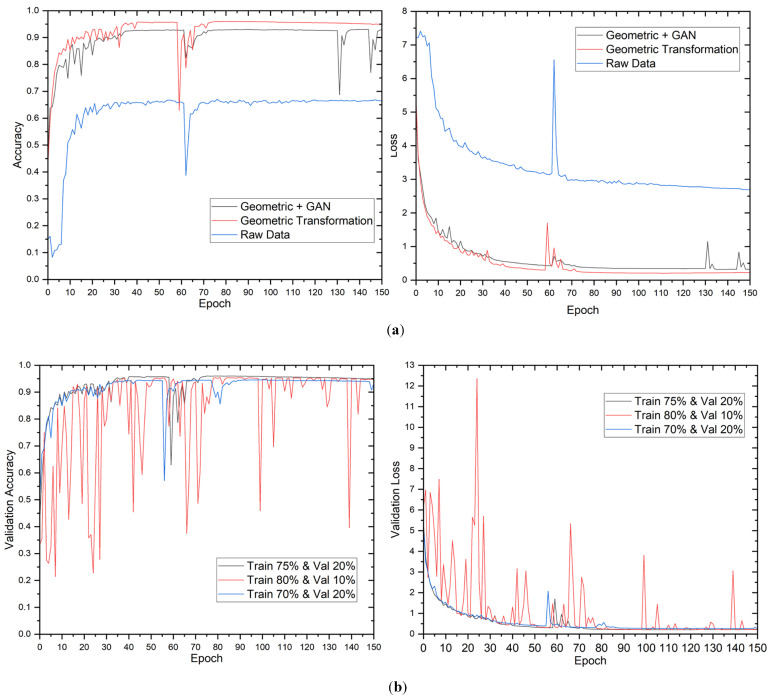
Evolution of validation loss and accuracy for 11-class outputs. Evolution of validation loss and accuracy in different (**a**) data augmentation methods and (**b**) data distributions.

**Figure 7 sensors-23-04918-f007:**
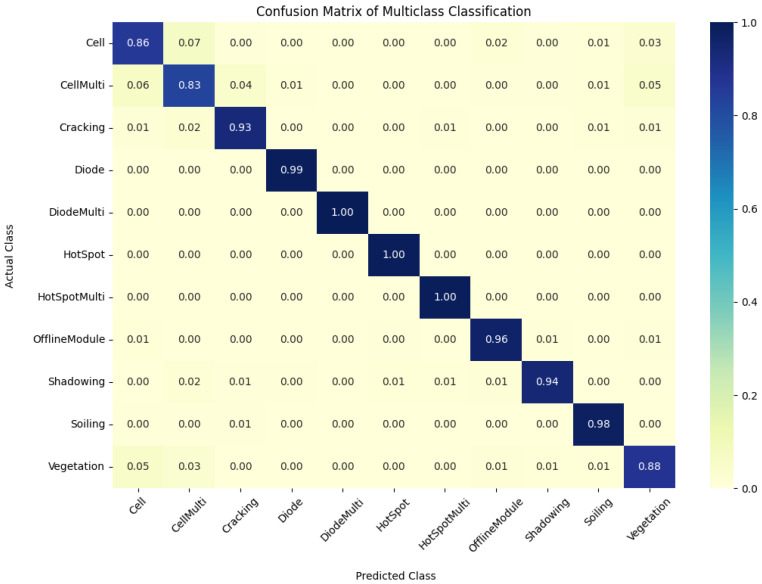
A confusion matrix of the coupled UDenseNet model for 11-class output.

**Figure 8 sensors-23-04918-f008:**
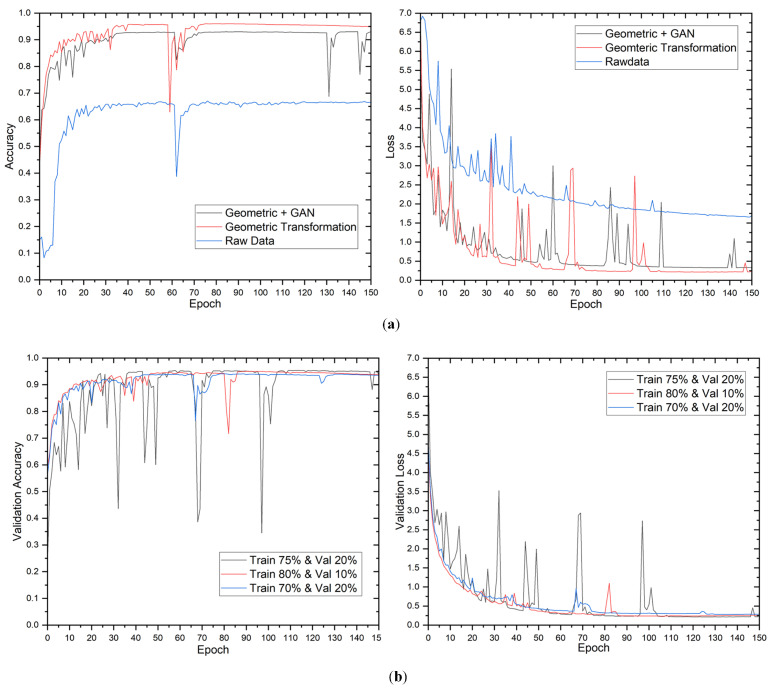
Evolution of validation loss and accuracy for 12-class outputs. Evolution of validation loss and accuracy in different (**a**) data augmentation methods and (**b**) data distributions.

**Figure 9 sensors-23-04918-f009:**
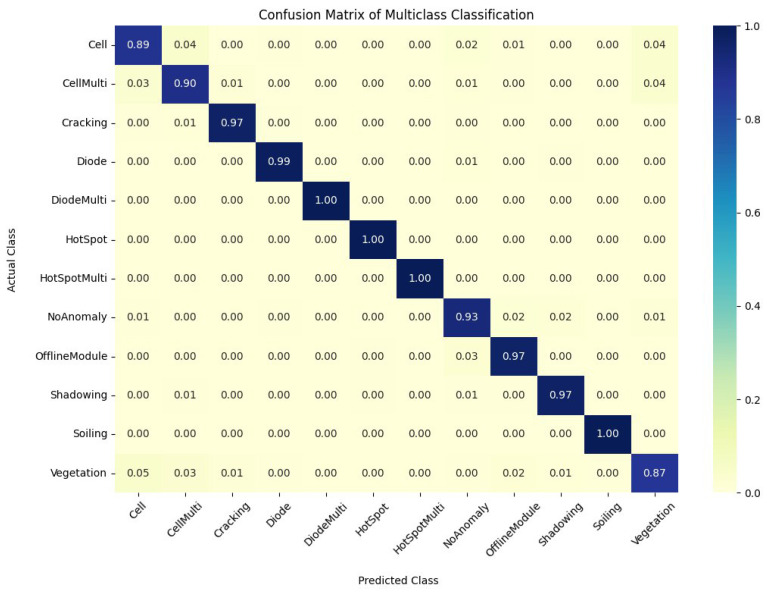
A confusion matrix of the coupled UDenseNet model for 12-class outputs.

**Table 1 sensors-23-04918-t001:** A comprehensive description and sample images of each class from dataset.

Class	Total Images	Description	Samples
Diode-Multi	175	Multiple activated bypass diodes, typically 2/3 of module.	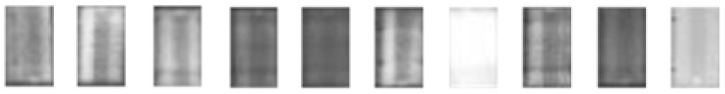
Soiling	204	Dust, dirt, or other debris on surface	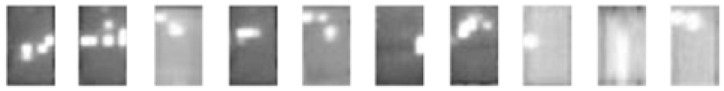
Hotspot-Multi	246	Multiple hot spots on a thin-film module.	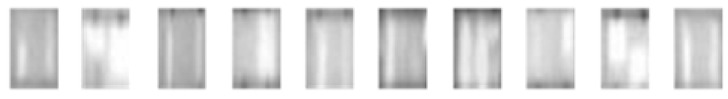
Hotspot	249	Hot spot on thin-film module.	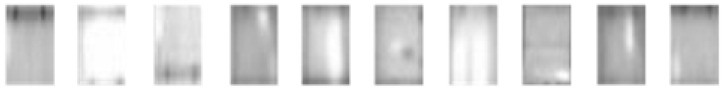
Offline Module	827	The entire module is heated.	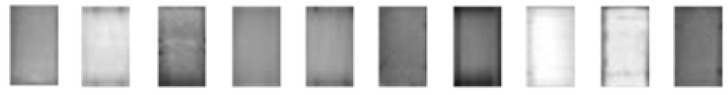
Cracking	940	Module anomaly caused by cracking on the module surface.	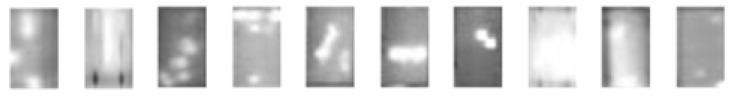
Shadowing	1056	Sunlight obstructed by vegetation, man-made structures, or adjacent rows.	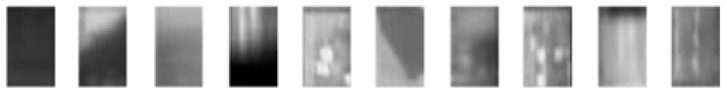
Cell-Multi	1288	Hot spots occurring with square geometry in multiple cells.	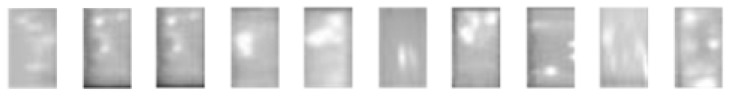
Diode	1499	Activated bypass diode,typically 1/3 of the module.	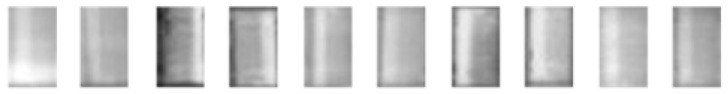
Vegetation	1639	Panels blocked by vegetation.	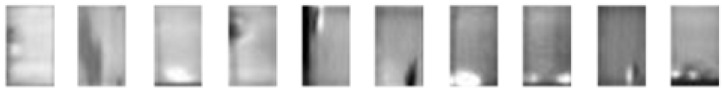
Cell	1877	Hot spot occurring with square geometry in single cell.	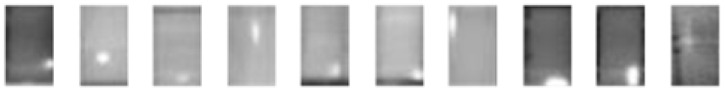
No Anomaly	10,000	Nominal solar module	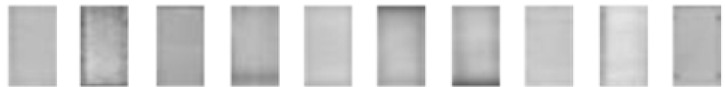
**Total**	**20,000**		

**Table 2 sensors-23-04918-t002:** Comparison of SSIM scores for raw and GAN-augmented images across different PV anomaly classes.

Classes	Raw Data			GAN	
	Max	Mean		Max	Mean
Diode	1.0	0.5558		0.8398	0.6907
Diode Multi	0.9479	0.4884		0.8399	0.6349
Hotspot	0.9449	0.6275		0.8497	0.7447
Hotspot Multi	0.9363	0.5217		0.7889	0.6528
Offline Module	1.0	0.6306		0.8686	0.7492
Soiling	0.9556	0.5223		0.6755	0.6407
Vegetation	1.0	0.6099		0.8443	0.7266
No Anomaly	0.9922	0.6548		0.8568	0.7874
Cell	0.9724	0.6211		0.8571	0.7376
Cell Multi	0.9503	0.5268		0.7985	0.6537
Cracking	0.8853	0.3941		0.6999	0.5261
Shadowing	1.0	0.5095		0.7797	0.6407

**Table 3 sensors-23-04918-t003:** Data distribution across training, validation, and testing sets for 2-class, 11-class, and 12-class experiments.

Data Augmentation	Training Set	Validation Set	Testing Set	Total Number of Images
**The First Case: Binary Classification**				
Raw Data	16,000 (80%)	2000 (10%)	2000 (10%)	
	15,000 (75%)	4000 (20%)	1000 (5%)	20,000 (100%)
	14,000 (70%)	4000 (20%)	2000 (10%)	
Geometric Transformation	70,400 (80%)	8800 (10%)	8800 (10%)	
	66,000 (75%)	17,600 (20%)	4400 (5%)	88,000 (100%)
	61,600 (70%)	17,600 (20%)	8800 (10%)	
Geometric + GAN	35,200 + 35,200 (GAN)	8800 (10%)	8800 (10%)	
	33,000 + 33,000 (GAN)	17,600 (20%)	4400 (5%)	88,000 (100%)
	31,300 + 31,300 (GAN)	17,600 (20%)	8800 (10%)	
**The Second Case: 11 Class Output**				
Raw Data	8000 (80%)	1000 (10%)	1000 (10%)	
	7500 (75%)	2000 (20%)	500 (5%)	10,000 (100%)
	7000 (70%)	2000 (20%)	1000 (10%)	
Geometric Transformation	88,000 (80%)	11,000 (10%)	11,000 (10%)	
	82,500 (75%)	22,000 (20%)	5500 (5%)	110,000 (100%)
	77,000 (70%)	22,000 (20%)	11,000 (10%)	
Geometric + GAN	44,000 + 44,000 (GAN)	11,000 (10%)	11,000 (10%)	
	41,250 + 41,250 (GAN)	22,000 (20%)	5500 (5%)	110,000 (100%)
	38,500 + 38,500 (GAN)	22,000 (20%)	11,000 (10%)	
**The Third Case: 12 Class Output**				
Raw Data	16,000 (80%)	2000 (10%)	2000 (10%)	
	15,000 (75%)	4000 (20%)	1000 (5%)	20,000 (100%)
	14,000 (70%)	4000 (20%)	2000 (10%)	
Geometric Transformation	96,000 (80%)	12,000 (10%)	12,000 (10%)	
	90,000 (75%)	24,000 (20%)	6000 (5%)	120,000 (100%)
	84,000 (70%)	24,000 (20%)	12,000 (10%)	
Geometric + GAN	48,000 + 48,000 (GAN)	12,000 (10%)	12,000 (10%)	
	45,000 + 45,000 (GAN)	24,000 (20%)	6000 (5%)	120,000 (100%)
	42,000 + 42,000 (GAN)	24,000 (20%)	12,000 (10%)	

**Table 4 sensors-23-04918-t004:** Hyperparameter tuning.

Hyperparameter	Setting Value
**Network hyperparameter**	
Batch size	32
Learning rate	0.001
Optimizer	AdamW [28]
Learning rate schedule	Monitoring: validation accuracy, Patience: 10
# of epochs	200
Early stopping	Monitoring: validation accuracy, Patience: 20
Loss function	[Binary cross entropy, Categorical cross entropy]
**Dense block hyperparameter**	
# of layer (L) for downsampling	2,3,6
# of layer (L) for upsampling	6,3,2
Spatial dropout rate	0.2

**Table 5 sensors-23-04918-t005:** The classification performance results of the coupled UDenseNet model for 2-class output.

Data Augmentation	Data Distribution (%)				Validation (%)				Test (%)			
	Train	Val	Test		Accuracy	Precision	Recall		Accuracy	Precision	Recall	F1
Raw Data	70	20	10		93.4	90.41	97.1		92.22	89.22	95.99	92
	75	20	5		93.12	90.36	96.55		92.63	89.88	95.99	93
	80	10	10		92.8	90.3	97.1		93.91	91.97	96.2	94
Geometric Transformation	70	20	10		99.34	98.69	**100**		99.17	98.37	**100**	99.12
	75	20	5		**99.61**	**99.22**	**100**		**99.39**	**98.79**	**100**	**99.4**
	80	10	10		99.22	98.46	**100**		99.25	98.52	**100**	99.18
Geometric + GAN	70	20	10		97.84	97.13	98.58		97.36	97.02	97.73	97.36
	75	20	5		97.72	97.19	98.27		96.93	96.19	97.7	96.93
	80	10	10		96.89	96.84	96.93		97.47	97.18	97.77	97.47

Note: the best performance for each metric in **bold**.

**Table 6 sensors-23-04918-t006:** The classification performance results of the coupled UDenseNet model for 11-class output.

Data Augmentation	Data Distribution (%)				Validation (%)				Test (%)			
	Train	Val	Test		Accuracy	Precision	Recall		Accuracy	Precision	Recall	F1
Raw Data	70	20	10		67.1	65.6	64.8		67.9	69.8	66.9	67
	75	20	5		68.4	66.8	65.7		64	65.1	63.3	64
	80	10	10		66	64.6	63		65.7	67.6	64.4	65
Geometric Transformation	70	20	10		94.6	94.92	94.36		94.43	94.86	94.3	94
	75	20	5		**95.99**	**96.22**	**95.83**		**96.65**	**96.75**	**96.61**	**97**
	80	10	10		95.5	95.8	95.3		95.1	95.4	94.8	95
Geometric + GAN	70	20	10		90.8	91.33	90.45		88.48	89.19	88.13	88.5
	75	20	5		93.39	93.89	93.19		94.22	94.78	93.97	94.2
	80	10	10		89.84	90.62	89.42		89.48	90.27	89.15	89.5

Note: the best performance for each metric in **bold**.

**Table 7 sensors-23-04918-t007:** The classification performance results of the coupled UDenseNet model for 12-class output.

Data Augmentation	Data Distribution (%)				Validation (%)				Test (%)			
	Train	Val	Test		Accuracy	Precision	Recall		Accuracy	Precision	Recall	F1
Raw Data	70	20	10		79.4	80.4	78.6		79.7	80.5	79.1	64
	75	20	5		78.9	79.9	78.3		78.6	79.5	78	59
	80	10	10		78.86	79.8	78.4		80.4	81.3	79.8	65
Geometric Transformation	70	20	10		94.1	94.5	93.9		94.2	94.6	93.9	94
	75	20	5		**95.3**	**95.6**	**95.1**		**95.7**	**96**	**95.5**	**96**
	80	10	10		95.1	95.4	94.8		94.7	95.2	94.5	95
Geometric + GAN	70	20	10		90.54	81.31	90.1		87.73	88.54	87.2	88
	75	20	5		92.58	93.19	92.24		93	93.51	92.9	93
	80	10	10		89.16	90	88.73		89	89.69	88.6	89

Note: the best performance for each metric in **bold**.

**Table 8 sensors-23-04918-t008:** The comparison of classification performance results of the related work in the same dataset for binary classification.

Model	Year	No. of Class	Total Parameter	Accuracy	Precision	Recall	F1
CNN [13]	2021	2	-	92.5%	92.00%	92.00%	-
Ensemble Model [12]	2021	2	**1.5M**	94.40%	-	-	-
Transfer Learning and Multiscale CNN [14]	2022	2	42M	97.32%	97.63%	97.00%	97.32%
**Proposed Method**	2023	2	13.9M	**99.39%**	**98.79%**	**100%**	**99.39%**

Note: the best performance for each metric in **bold**.

**Table 9 sensors-23-04918-t009:** The comparison of classification performance results of the related work in the same dataset for multi-class classification.

Model	Year	No. of Class	Total Parameter	Accuracy	Precision	Recall	F1
CNN [13]	2021	11	-	66.43%	-	-	-
Ensemble Model [12]	2021	12	**1.5M**	85.9%	-	-	-
Transfer Learning and Multiscale CNN [14]	2022	11	42M	93.51%	93.52%	93.51%	93.49%
**Proposed Method**	2023	11	13.9M	**96.65%**	**96.75%**	**96.61%**	**97.00%**
		12	13.9M	95.72%	96.01%	95.53%	97.00%

Note: the best performance for each metric in **bold**.

## Data Availability

The Infrared Solar Modules dataset by Raptor Maps is a third party dataset accessible at: https://github.com/RaptorMaps/InfraredSolarModules (last accessed 21 February 2023).

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
