# Peer review of "A Novel Approach for Efficient Solar Panel Fault Classification Using Coupled UDenseNet"

_sensors, 2023, doi:10.3390/s23104918_

Round 1

Reviewer 1 Report

1.    Overview:

This paper proposes a lightweight coupled UdenseNet model for accurately detecting and classifying faults in photovoltaic (PV) systems from aerial thermography images. The proposed model achieves high accuracy and efficiency, making it suitable for real-time analysis of large-scale solar farms. The use of geometric transformation and generative adversarial networks (GAN) image augmentation techniques is also found to improve the model's performance.

The article has clear logic and detailed experiments. However, there are still some minor issues, and I suggest that the article be accepted after minor revisions.

2. Minor modification problems:

(1) The title of figure 3 is incorrect, as it includes ”(a)” unnecessarily..

(2) The abstract of the article mentions that "The proposed model was also found to be more efficient in terms of parameter counts and processing time," but there was no test conducted on the processing time of the model in the experimental section.

(3) In the section introducing evaluation metrics, a brief overview of evaluation metrics for model parameters should be provided.

Author Response

Dear Reviewer,

We would like to thank you for the comments and the opportunity to resubmit a revised version. 
We have updated our manuscript according to your advice and highlighted it in yellow color. 
Please see the attachment.
Thank you very much.

Best regards,

Professor Yeong Min Jang
Department of Electronics Engineering,
Kookmin University, Seoul, Korea.

Reviewer 2 Report

The frequent inspection and maintenance of photovoltaic (PV) systems are always needed to predict malfunctions, safety issues, elongate the PV module lifespan, and prevent fire hazards, and heavy metal-containing PV module waste. In this contribution, the authors present a novel approach to detect the solar panel fault using coupled UDense-Net, which showed distinctly superior to the previously established methods. This is an interesting and comprehensive work and the manuscript is well-written. I would like to recommend it for publication in Sensors after minor revision:

1)     This study is mainly based on theoretical predications. Have the authors utilize their method to analysis the real PV modules with and without defects to make a comparison study to verify the effectiveness of the proposed method.

2)     In the Experimental section, the providers of any samples should be described, and the detailed information about the sample should be provided.  

This manuscript is well-written. 

Author Response

(The authors gave the same response as above.)

Reviewer 3 Report

The authors analysed solar panel failures based on photos. Three models were considered, such as: 2, 11 and 12 failure classes.  The authors compared Deep Neural Network (DNN) structures and proposed an effective failure classification model, and considered different methods of preparing images for DNN training.

Please clarify:

-     -   What is the application of the failure classes used in the study? Which approach is more useful 2, 11 or 12 class model?

-    -    The various models used in the study have varying predictive accuracy, but how about the utility of the approaches presented?

 I have no comments.

Author Response

(The authors gave the same response as above.)

Reviewer 4 Report

This manuscript proposes a coupled UDenseNet method for classifying Solar Panel Faults, which outperforms previous studies. Overall, the manuscript is well-written, with thorough literature reviews and well-designed experiments. However, some improvements can be made:

In the abstract, it would be beneficial to summarize the methodologies of previous studies and mention their corresponding flaws to highlight the advantages of this work.

In the abstract, a clear definition of PV faults should be provided, along with examples of the 11 or 12 classes of faults that are being specified.

On page 2, it would be helpful to clarify the overall accuracy performance of the Electrical assessment.

On page 2, it would be better to introduce "photovoltaic (PV)" at the beginning of the main text.

On page 2, it would be clearer to point out that the 11 and 12 types of faults are introduced in Table 1.

Instead of "Related works", it would be better to use "Literature Review".

On page 4, it would be useful to specify which anomalies and others are being referred to.

On page 6, it would be helpful to clarify the motivation behind choosing this particular network combination.

Author Response

(The authors gave the same response as above.)
